# Complete Chloroplast Genome of Bamboo Species *Pleioblastus ovatoauritus* and Comparative Analysis of *Pleioblastus* from China and Japan

Weihan Peng, Beibei Wang, Zhuolong Shen and Qirong Guo *

Co-Innovation Center for Sustainable Forestry in Southern China, College of Forestry, Nanjing Forestry University, Nanjing 210037, China
* Correspondence: qrguo@njfu.edu.cn

**Abstract:** *Pleioblastus ovatoauritus* T.H.Wen ex W.Y.Zhang is bamboo species published in 2018, originated from and existing in southeastern China. The chloroplast genome of *Pl. ovatoauritus* was obtained using a high-throughput sequencing platform. The chloroplast genome is up to 139,708 bp in length and displays a typical quadripartite structure with one large single-copy region, one small single-copy region, and two inverted repeat regions. There are 82 protein-coding genes, 8 rRNA genes, and 39 tRNA genes in the plastome genome. However, the interspecific relationship of *Pleioblastus* species originated from China and Japan has not been revealed explicitly. To understand their relationship, data from four Chinese species and four Japanese species were selected to investigate the distinctions between their genome structures, codon usage patterns, and SSR sites. We moved forward to examine the sequence divergence and polymorphic sites between the eight species. Phylogenetic trees were then plotted using the maximum likelihood method based on different parts of the sequences. Obvious difference found in the JLB boundary and a split in the phylograms contributed to our decision to split *Pleioblastus* species of China and Japan into different clades. Moreover, taxonomy using the subgenera concept in *Flora Reipublicae Popularis Sinicae* proved untenable. Nine SSR primers for *Pleioblastus* genus were then developed from cp genomes, aimed at facilitating identification and germplasm investigation.

**Keywords:** *Pleioblastus ovatoauritus*; chloroplast genome; Chinese clade; Japanese clade; phylogenetic analysis; SSR primer

## 1. Introduction

Bambusoideae is a subfamily of Poeace with over 139 accepted genera and 1700 species [1], leaving its imprint in all major continents in temperate and tropical or woody or herbaceous climates [2]. The traditional method of classifying bamboo species based on their reproductive organs has limitations due to their irregular blossoming and fruiting habits. Recent studies also suggested reticulation and hybridization phenomena in their origin [3,4]. DNA sequencing and molecular markers such as SSR and SNP markers have been employed to resolve the phylogeny, population origin, genetic evolution, diversity, and phylogenetic identification of Poaceae species [5].

*Pleioblastus* is a tough genus of woody bamboo found mainly in eastern Asia. It is used by locals for the handles of brushes and umbrellas, and some species are used for gardening due to their beauty. In the flora of China, *Pleioblastus* species are first divided into two groups according to their height and leaf blades [6]. Dwarf species with leaf blades variegated or closely distichous are grouped and described as native to Japan. Species of normal height, not variegated, separated, and not distichous are grouped and described as endemic to China. In *Flora Reipublicae Popularis Sinicae* (Volume 9, part 1), the genus is included in Subtribe Arundinariinae with two subgenera, namely, Subgen. *Pleioblastus* and Subgen. *Nipponocalamus* [7]. Subgen. *Nipponocalamus* is further divided into Sect.

*Nipponocalamus* and Sect. *Amari* based on the dehiscence on the apical palea. Although both FOC and FRPS failed to include all *Pleioblastus* species, we were inspired by FOC to consider that the *Pleioblastus* classification may have strong connection with the origin places of species. According to Clark (2010) and Li (2012), *Pleioblastus* is a polyphyletic genus with its species separated in different clades, and the interspecific relationship remains to be reevaluated due to their mass distribution in different clades [8]. Meanwhile, the polyphyletic character of *Pseudosasa, Sasa*, and *Pleioblastus* posed challenges to developing molecular markers with a demand for higher resolution [8]. *Pleioblastus ovatoauritus* T.H. Wen ex W.Y. Zhang is a newly discovered species in China, found and originally named by Taihui Wen, which was then adopted and formally published by Yue Jin-jun [9]. With culm of 6–7 m, internodes of 30–45 cm, and culm sheaths of a greenish-yellow turning to yellowish-brown, the bamboo greatly resembles *Pl. maculatus* but without a setose ring in the base of the culm sheath (see Figure 1).

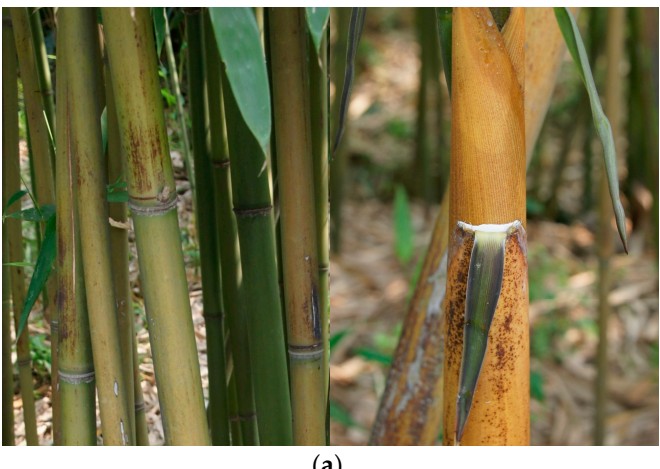

(**a**)

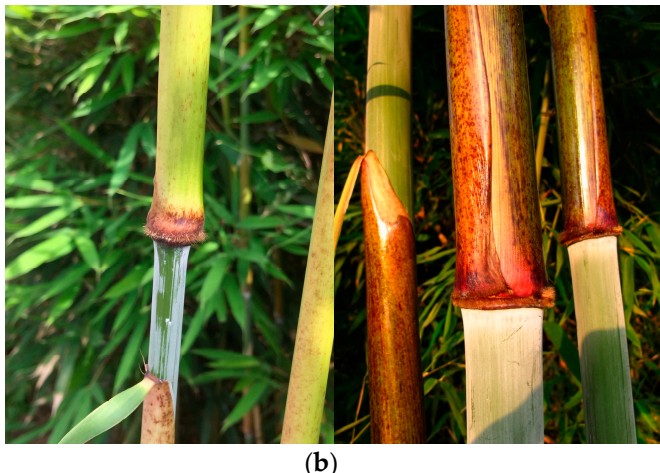

(**b**)

**Figure 1.** Base of culm sheaths on *Pl. ovatoauritus* and *Pl. maculatus*. (**a**) Naked base of the culm sheath on *Pl. ovatoauritus*. (**b**) Setose ring in the base of culm sheath on *Pl. maculatus*.

In plants, the chloroplast is a crucial organelle that performs photosynthesis and generates energy for the plant. It contains a circular DNA genome with a quadripartite structure consisting of one large single-copy (LSC) region, one small single-copy (SSC) region, and two inverted repeat (IR) regions [10]. The chloroplast genome spans 15,553 base pairs (bp) to 521 kbp [11], containing genes which function in photosynthesis or self-replication. The chloroplast genome is highly conserved, with no gene recombinations happening in the replication process of cells. Due to its high conservation, the chloroplast genome becomes a useful tool for studying phylogenetic relationships between different species [12–15].

In this study, we sequenced and annotated the chloroplast genome of *Pl. ovatoauritus*, the newly published species. Seven other *Pleioblastus* species' chloroplast genomes were also downloaded from NCBI to investigate the interspecific relationships within the genus *Pleioblastus*. Species in the Japanese group are all dwarf bamboos for ornamental usage, while Chinese group's bamboo grows to normal height. Codon usage pattern, IR boundary variation, polymorphic sites, and sequence divergence were examined with the aim of revealing the differences between these species. Phylogenetic trees were drawn at last, based on different parts of the genome. We aimed to figure out the phylogenetic relationship inside the *Pleioblastus* species and provide valid evidence for the pros or cons of the split view. Additionally, general SSR primers were developed for the eight species, which could be used in germplasm investigation and phylogenetic analysis of *Pleioblastus*.

## 2. Materials and Methods

### 2.1. DNA Sample and Data Collection

Fresh leaves sample of *Pl. ovatoauritus* were collected from living plants in the bamboo garden of Anji, Zhejiang province, China (30°38′ N, 119°41′ E), by Guo QR and Zhou Jie on 18 October 2018. Total DNA was extracted from 100 mg fresh leaves using Qiagen Plant Genomic DNA Prep Kit (Sangon Biotech, Shanghai, China) and, through agarose gel electrophoresis, we purified the DNA sample and sequenced it using Illumina Hiseq 2500 platform. A total of 37.8 gb original sequencing data were obtained. The voucher specimens are preserved in the Germplasm Gene Bank of the Co-Innovation Center for Sustainable Forestry in Southern China, Nanjing Forestry University.

Data of *Pl. amarus* (Keng) Keng f.; *Pl. maculatus* (McClure) C.D.Chu and C.S.Chao; *Pl. triangulata* (Hsueh and T.P.Yi) N.H.Xia, Y.H.Tong, and Z.Y.Niu; *Pl. argenteostriatus* (Regel) Nakai; *Pl. fortunei* (Van Houtte) Nakai; *Pl. pygmaeus* 'Disticha'; *Pl. pygmaeus* (Miq.) Nakai; *Phyllostachys reticulata* (Ruprecht) K. Koch; *Ph. edulis* (Carriere) J. Houzeau; *Ph. edulis* 'Yuanbao'; *Ph. propinqua* McClure; *Ph. violascens* (Carrière) Riviere and C. Rivière; *Shibataea chiangshanensis* T. W. Wen; and *S. kumasaca* (Zoll. ex Steud.) Makino ex Nakai were downloaded from NCBI (National Center for Biotechnology Information (nih.gov)). Details of their cp genomes are shown in Table 1. The scientific name of the material used in chloroplast genome accession MW874473, whose name was originally *Phyllostachys edulis* f. *tubaeformis*, was corrected to *Phyllostachys edulis* 'Yuanbao' according to the document and confirmation with the author, Yue Jin-Jun [16,17].

**Table 1.** The genera used in comparative analysis or phylogenetic analysis.

| Genus | Taxon | Accession |
|---|---|---|
| Ingroup | | |
| | *Pleioblastus ovatoauritus* T.H.Wen | OP235916 |
| | *Pl. maculatus* (McClure) C.D.Chu and C.S.Chao | JX513424 |
| | *Pl. amarus* (Keng) Keng f. | NC043892 |
| *Pleioblastus* | *Pl. ortuneate* (Hsueh and T.P.Yi) N.H.Xia, Y.H.Tong and Z.Y.Niu | OK323193 |
| | *Pl. argenteostriatus* (Regel) Nakai | OP036432 |
| | *Pl. ortune* (Van Houtte) Nakai | OP036433 |
| | *Pl. pygmaeus* 'Disticha' | OP036434 |
| | *Pl. pygmaeus* (Miq.) Nakai | OP036435 |
| Outgroup | | |
| | *Phyllostachys edulis* (Carriere) J. Houzeau | MW007170 |
| | *Ph. Edulis* 'Yuanbao' | MW874473 |
| *Phyllostachys* | *Ph. Reticulata* (Ruprecht) K. Koch | MN537808 |
| | *Ph. Propinqua* McClure | JN415113 |
| | *Ph. Violascens* (Carrière) Riviere and C. Rivière | OP612331 |
| *Shibataea* | *Shibataea chiangshanensis* T. W. Wen | NC036826 |
| | *S. kumasaca* (Zoll. Ex Steud.) Makino ex Nakai | KU523578 |

*Pl. amarus*, *Pl. macualtus*, and *Pl. triangulata* have the same origin as *Pl. ovatoauritus*, growing in mainland China. These species are of normal culm length and their leaf blades are not variegated, separated, and not distichous. *Pl. argenteostriatus*, *Pl. fortunei*, *Pl. pygmaeus* 'Disticha', and *Pl. pygmaeus* are all dwarf species that are native to Japan. *Phyllostachys* is a genus with plenty of well-known species and the relationship concerning *Phyllostachys* is clearer. Compared to some of Arundinarieae genera, such as *Indosasa* or *Pseudosasa*, *Phyllostachys* species tend not to mix with other genera when clustering, thus providing evident reflection in analysis. *Shibataea* also has advantages similar to *Phyllostachys*, but has a closer relationship with *Phyllostachys*.

*2.2. Chloroplast Genome Assembly and Annotation*

Raw data were assembled using GetOrganelle [18] with primitive parameter kmer = 85, and *Pl. amarus* was selected as the seed. The primary sequence was examined using Mauve [19] to find out if there were any errors in assembly. The cp genome was uploaded to the website application GeSeq (https://chlorobox.mpimp-golm.mpg.de (accessed on 15 August 2022)) for annotation assistance with manual correction by Genious (Version 8.0.1) [20]. Chloroplot [21] was employed to visualize the chloroplast genome. The annotated cp genome was uploaded to NCBI and was successfully adopted with the Genbank ID: OP239516.

*2.3. Comparative Analysis of Complete Genomes*

Before researching details of the genomes' structures, Mauve was again used to investigate if there were any assembling errors and to find out if there were inversion or disorder regions on all the 8 species' genomes. IRscope [22] was used for genome structure analysis, mainly on the extension or contraction of IRs of the genome.

We aligned 10 species by MAFFT v7.505 [23] in order to implement microstructural mutations and sequence polymorphism analysis using Dnasp 6.0 [24]. Total sequence, LSC, SSC, IR, and CDS sequences will be analyzed, respectively, by the sliding window method. For the better visualization of the DNA diversity, mVISTA [25] was then employed using the Shuffle-LAGAN model to produce cross-sectional comparison graph.

For the better understanding of the expression of chloroplast plastome genes and the internal phylogenic process of the 8 species in *Pleioblastus* genus, CodonW (Version 1.3) [26] was used for counting and analyzing the CUS and RSCU values of 82 genes that were common in all 8 species. CAI, CBI, and GC3 values were also involved to predict the usage bias of each genome.

*2.4. Phylogenic Analysis*

In order to obtain comprehensive and more accurate results, full sequences, LSC regions, SSC regions, single copy regions, and 78 CDS sequences of 15 species (8 *Pleioblastus*, 5 *Phyllostachys,* and 2 *Shibataea*) were selected for the phylogenic analysis. *Ph. reticulata*, *Ph. violasens*, *Ph. propinqua*, *Phyllostachys edulis*, *Phyllostachys edulis* 'Yuanbao', *S. chiangshanensis* and *S. kumasaca* were selected as outgroup. A total of 10 species' Sequences were aligned by MAFFT. MEGA 11 was used for the final phylogenic and evolutionary history analyzing by the maximum likelihood method [27]. After 1000 repetitions, bootstrap consensus trees were inferred. Interactive Tree Of Life (iTOL) v5 was used for the polishing of trees [28].

*2.5. Development of Primers for SSRs*

We used the website, MISA, for the positioning of SSRs in 8 species [29]. The minimum repeating times of mono nucleotide repeats, dinucleotides repeat, trinucleotides, tetranucleotides repeats, and pentanuclotides repeats were required up to 9, 5, 4, 4, 4, respectively. Primer 3 Plus [30] was employed for the development of primers.

**3. Results**

*3.1. Chloroplast Genome of Pl. ovatoauritus*

Based on the Illumina Hiseq 2500 platform, raw sequencing data of two directions obtained from *Pl. ovatoauritus* reach 3174–3209 MB and were loaded to GetOrganelle. Trimming and choosing reads was unnecessary because GetOrganelle would estimate and trim reads autonomously as well as the Kmer value decision and the word-size of the reads. Actually, 30,000,000 reads (15,000,000 + 15,000,000) were used with the word-size of 102 bp. After 10 rounds of extension, three Kmer values were chosen (21, 65 and 105) to estimate the length of cp genome. Additionally, *Pl. amarus* (Genbankid:NC043892) was selected to provide the seeds. The assembly result reaches 139,708 bp in length and was uploaded to website application Geseq (MPI-MP CHLOROBOX-GeSeq (mpg.de)) for annotation. After

manually trimming in Geneious, wrong and extra annotations are corrected. The circular DNA was then visualized (Figure 2). The cp genome and its annotation were uploaded to NCBI. (Genbankid:OP235916).

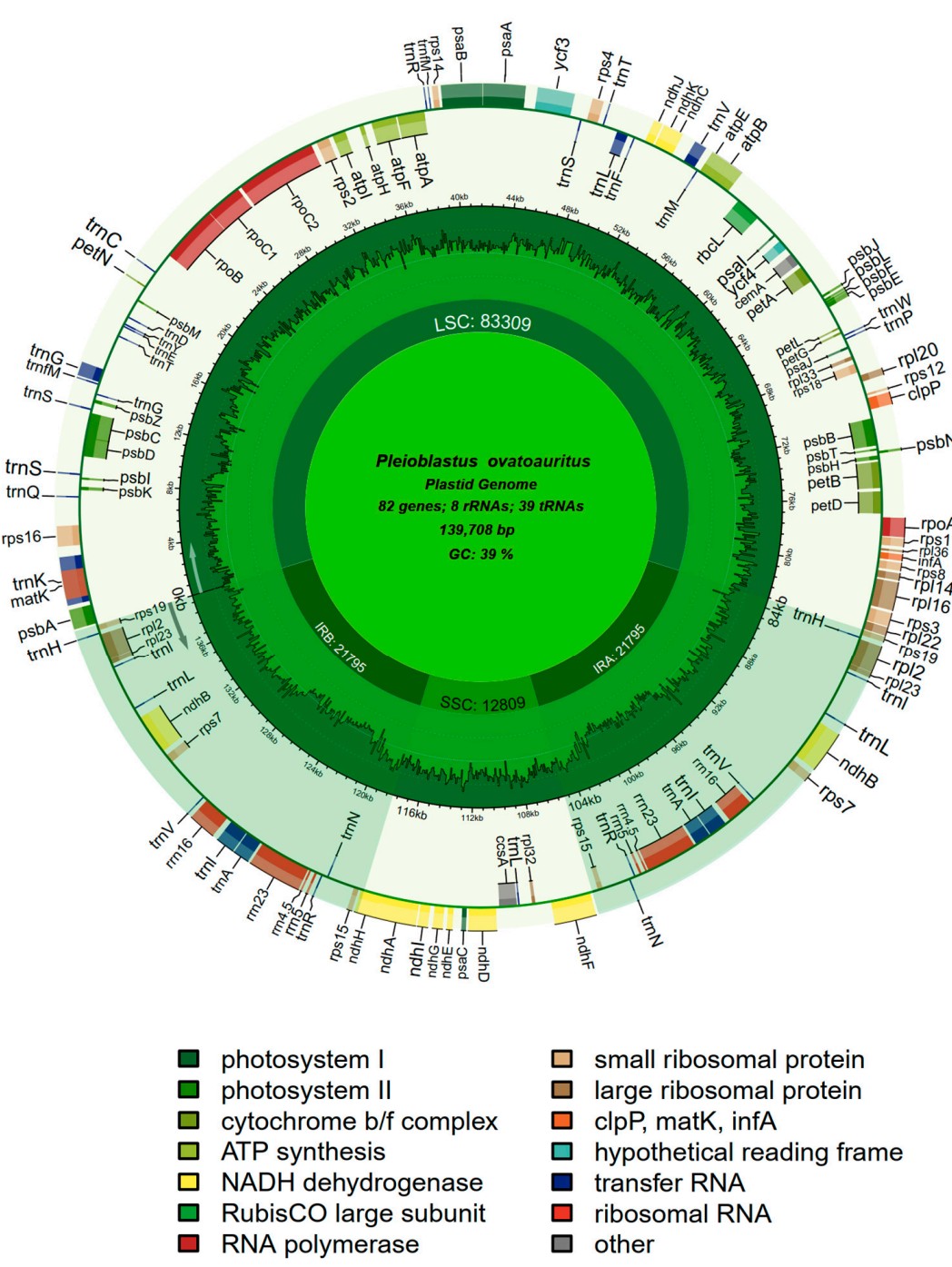

**Figure 2.** Gene map of *Pl. ovatoauritus*. The outer circle shows positions of genes and the blocks with different colors represent different kinds of gene. The inner circle shows the tetrad structure of the genome; the dark green shadow bar shows the GC content.

The plastome genome of *Pl. ovatoauritus* shares a typical quadripartite structure with other bamboos (i.e., *Phyllostachys*, *Bambusa*) that contain one LSC, one SSC, and a pair of IRs (Table 2). The lengths of LSC, SSC, and single IR are 83,309 bp (59.6%), 12,809 bp (9.2%), and 21,795 bp (15.6%), respectively, while length of the gene coding sequences reaches 59,322 bp

(42.5%). Furthermore, the GC content of each region comes to 37.0%, 33.4%, 44.2%, and 39.4%. The results above are in line with those found in other species of *Pleioblastus* [31].

**Table 2.** Chloroplast genome information of eight *Pleioblastus* species.

| Taxon | *Pl. ovatoauritus* | *Pl. maculata* | *Pl. amarus* | *Pl. triangulata* | *Pl. argenteostriatus* | *Pl. fortunei* | *Pl. pygmaeus* 'Disticha' | *Pl. pygmaeus* |
|---|---|---|---|---|---|---|---|---|
| Accession No. | OP235916 | JX513424 | NC043892 | OK323193 | OP036432 | OP036433 | OP036434 | OP036435 |
| Plastome legnth | 139,708 | 139,720 | 139,703 | 139,690 | 139,031 | 139,067 | 139,067 | 139,032 |
| LSC | 83,309 | 83,283 | 83,265 | 83,283 | 82,579 | 82,587 | 82,587 | 82,580 |
| SSC | 12,809 | 12,847 | 12,846 | 12,817 | 12,860 | 12,888 | 12,888 | 12,860 |
| IR (single) | 21,795 | 21,795 | 21,796 | 21,795 | 21,796 | 21,796 | 21,796 | 21,796 |
| GC content % | 38.9 | 38.9 | 38.9 | 38.9 | 38.9 | 38.9 | 38.9 | 38.9 |
| No. of CDS | 82 | 82 | 82 | 82 | 82 | 82 | 82 | 82 |
| No. of tRNAs | 39 | 39 | 39 | 36 | 30 | 30 | 30 | 30 |
| No. of rRNAs | 8 | 8 | 8 | 8 | 8 | 8 | 8 | 8 |

There are 129 genes contained in the cp genome of *Pl. ovatoauritus*, including 82 protein coding genes, 39 tRNA genes, and 8 rRNA genes (Table 3). Among all the genes, 81 of them are located in LSC, 10 are located in SSC, and 36 are located in IRs. The remaining two genes are *rps12* and *ndhH*; *rps12* is a cross-border gene with one part in LSC and the other part in IR region, while *ndhH* crosses the boundary between the SSC region and the IR region. There are 15 photosystem II genes, 12 NADH dehydrogenase genes, 6 Cytochrome b/f complex genes, 6 ATP synthase genes, 5 photosystem I genes, 1 Rubisco large subunit gene, and 1 C-type cytochrome synthesis gene which participate in photosynthesis. A total of 15 ribosomal proteins genes, 11 ribosomal proteins genes, 4 RNA polymerase genes, 8 rRNA genes, 1 maturase gene, and 39 tRNA genes are involved in self-replication. Additionally, *cemA* gene encodes the chloroplast envelope membrane protein, *clpP* gene helps construct ATP-dependent protease, and *infA* gene is associated with the translational initiation factor. The remaining two genes are *ycf3* and *ycf4*, whose functions are unknown. Notably, gene *ycf68*, which also belongs to hypothetical genes, appears with an advanced stop codon in its theoretical region in *Pl. ovatoauritus*, *Pl. maculatus*, and *Pl. triangulate*. The unexpected stop codon leads to a half cut of the gene compared to that of *Pl. amarus*, and we decided to label it as a pseudogene and remove it from the annotation.

**Table 3.** All genes in different groups and systems in *Pl. ovatoauritus*.

| System | Group | Name |
|---|---|---|
| Photosynthesis | Subunits of ATP synthase | *atpA, atpB, atpE, atpF, atpH, atpI* |
| | Subunits of NADH-dehydrogenase | *ndhA, ndhB<sup>a</sup>, ndhC, ndhD, ndhE, ndhF, ndhG, ndhH, ndhI, ndhJ, ndhK* |
| | Subunits of cytochrome b/f complex | *petA, petB, petD, petG, petL, petN* |
| | Subunits of photosystem I | *psaA, psaB, psaC, psaI, psaJ* |
| | Subunits of photosystem II | *psbA, psbB, psbC, psbD, psbE, psbF, psbH, psbI, psbJ, psbK, psbL, psbM, psbN, psbT, psbZ* |
| | Subunit of rubisco | *rbcL* |
| Transcription and translation | Large subunit of ribosome | *rpl2 [a], rpl14, rpl16, rpl20, rpl22, rpl23 [a], rpl32, rpl33, rpl36* |
| | DNA dependent RNA polymerase | *rpoA, rpoB, rpoC1, rpoC2* |
| | Small subunit of ribosomal proteins | *rps2, rps3, rps4, rps7 [a], rps8, rps11, rps12, rps14, rps15 [a], rps16, rps18, rps19 [a]* |
| | rRNA genes | *rrn4.5 [a], rrn5 [a], rrn16 [a], rrn23 [a]* |
| | tRNA genes | *trnA-UGC [a], trnC-GCA, trnD-GUC, trnE-UUC, trnF-GAA, trnfM-CAU [a], trnG-GCC, trnG-UCC, trnH-GUG [a], trnI-CAU [a], trnIGAU [a], trnK-UUU, trnL-CAA [a], trnL-UAA, trnL-UAG, trnM-CAU, trnN-GUU [a], trnP-GGG, trnQ-UUG, trnR-ACG [a], trnR-UCU, trnS-GCU, trnS-GGA, trnS-UGA, trnT-GGU, trnT-UGU, trnV-GAC [a], trnV-UAC, trnW-CCA, trnY-GUA* |
| Other genes | c-type cytochrome synthesis gene | *ccsA* |
| | Envelope membrane protein | *cemA* |
| | ATP-dependent protease | *clpP* |
| | Maturase | *matK* |
| | Hypothetical chloroplast reading frames | *ycf3, ycf4* |
| | Translational initiation factor | *infA* |

"*a*"——gene with two copies.

### 3.2. Chloroplast Genome Structure Analysis of Genus Pleioblastus

We use the Mavue plugin in Geneious to implement colinearity analysis for the eight species of *Pleioblastus*, and the result indicated that there is no inversion or reordering in the cp genome of *Pl. ovatoauritus*, as well as in other *Pleioblastus* genera. Then, we uploaded genome data to online application IRscope (https://irscope.shinyapps.io/irapp/ (accessed on 20 August 2022)) to compare and analyze the difference of junctions between repeat regions and single copy regions within the eight cp genomes (Figure 3). Though cp genomes have comparatively conversed structure, there were still some variations like extension or contraction in the junction of different sectors. Genes *rpl22* in the LSC of eight species are close to the border between IRb and LSC, with the same distance of 24 bp. Gene *rps19* in both IR regions have the gap of 42 bp towards the junction. Notably, all the IRa regions in *Pleioblastus* genus have extended to the gene *ndhH* at 187 bp. The most significant difference lies in the distance from the end of *ndhF* to the JLB, where populations from China have shorter distance than group from Japan. *Pl. ovatoauritus* and *Pl. triangulata* have a distance of 142 bp to the IRb; *Pl. amarus* and *Pl. maculata* have 155 bp. In the Japanese group, the distances increase to 196 bp in *Pl. pygmaeus* and *Pl. argenteostriatus* and 233 bp in *Pl. fortunei* and *Pl. pygmaeus* 'Disticha'. Resultingly, the conspicuous difference in the distance between *ndhF* and JLB corroborates the divergence of two groups to some extent.

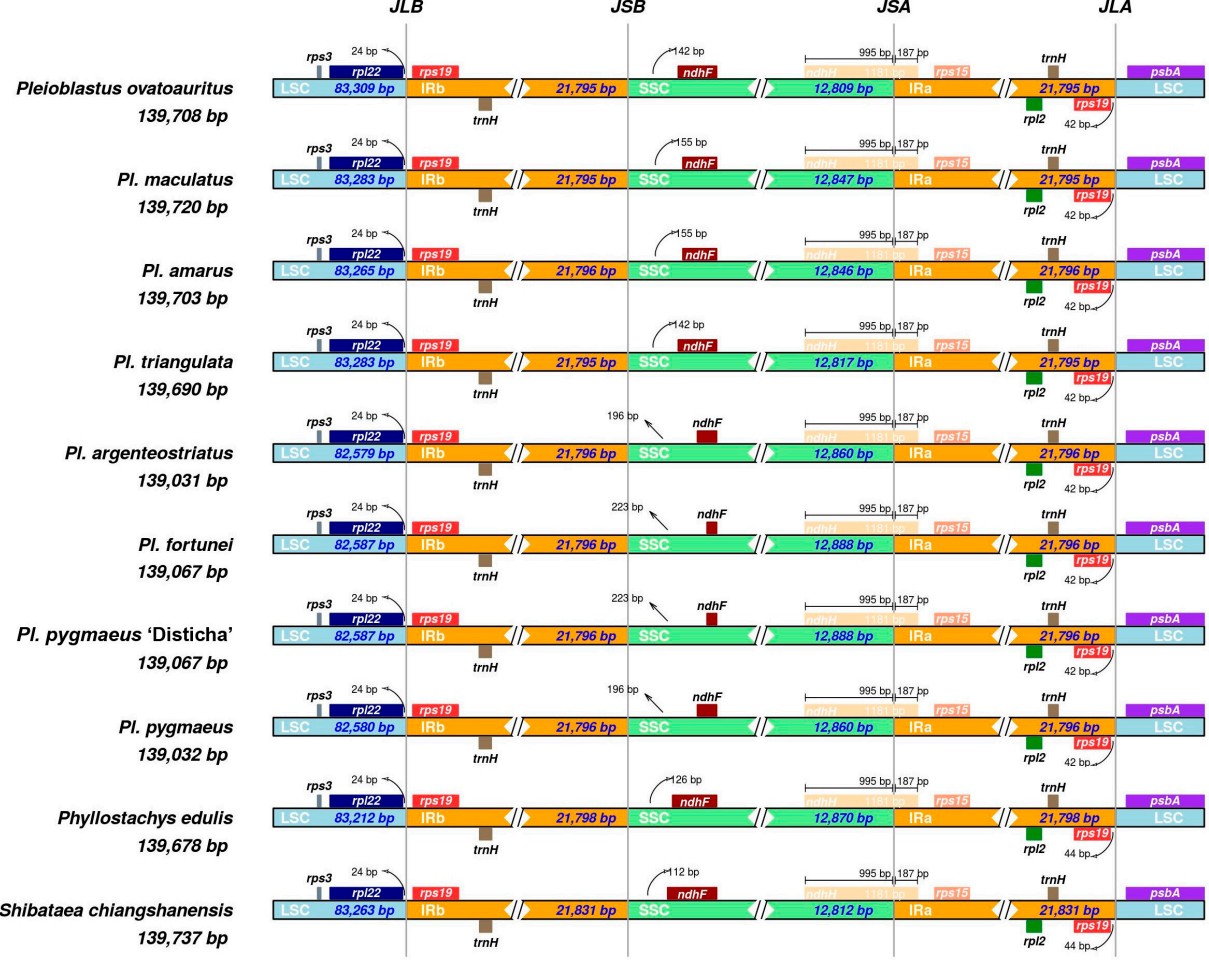

**Figure 3.** Chloroplast genome boundary comparative analysis of eight species by IRscope. JLB/JLA means the junction between LSC and IRb/Ira, and JSB/JSA means the junction between SSC and IRb/IRa.

### 3.3. Codon Usage

A total of 82 genes' coding sequences were involved in the analysis (*accD*, *ycf1*, *ycf2*, *ycf15*, and *ycf68* of some species were excluded for uniformity). We first detected the start codons, which normally appear to be 'AUG' (coding Methionine). A total of 80 coding sequences start with the 'M' codon, while *rp12* starts with 'I' codon or 'T' codon (AUA or ACG, coding l-isoleucine, or L-Threonine) and rps19 starts with 'V' codon (GUG, coding Valine). In the Chinese group, *rp12* starts with 'T' codon while the Japanese group's starts with 'I' codon. Then, RSCU values were obtained to compare the codon pattern of four genera. Together with RSCU, analysis of optimal codons with specific data are listed in Table 3, some basic statistics are also included. The statistics below indicate that eight *Pleioblastus* species possess similar codon usage patterns, with only delicate difference (Table 4). pleucine coding codons account for most of the codons up to 10.7% while cysteine coding codons are the least in eight bamboos, taking up only 1.1%. Additionally, the Chinese group tends to have higher GC content in CDS but less protein-coding codons. RSCU values would be important in evolutional analysis because they reflect the changes of the codon usage frequency. The third position of a codon tends to be the first position undergoing changes, and the changes will cause less alternation of the protein it codes. Therefore, we involved the starts of GC3, A3s, T3s, etc. *Pl. maculatus* and *Pl. triangulata* have 30 codons compared to others (29 codons). The cause of changes lies in the tiny increase of the number of the codon 'UCA' (coding Serine), from 1.00 to 1.01.

**Table 4.** Codon usage situation, preferred codons, and optimal codons of *Pleioblastus* genus.

| Genus | | *Pl. ovatoauritus* | *Pl. amarus* | *Pl. maculata* | *Pl. triangulata* | *Pl. argenteostriatus* | *Pl. fortunei* | *Pl. pygmaeus 'Disticha'* | *Pl. pygmaeus* |
|---|---|---|---|---|---|---|---|---|---|
| GC content of CDS (%) | | 39.5 | 39.5 | 39.5 | 39.5 | 39.4 | 39.4 | 39.4 | 39.4 |
| Codon preference at 3rd position | | T (45.3%) | T (45.3%) | T (45.3%) | T (45.3%) | T (45.3%) | T (45.3%) | T (45.3%) | T (45.3%) |
| Total AA | | 19,695 | 19,695 | 19,695 | 19,695 | 19,705 | 19,705 | 19,705 | 19,705 |
| Most preferred stop codon | | UAA | UAA | UAA | UAA | UAA | UAA | UAA | UAA |
| Most frequent AA | | Leu | Leu | Leu | Leu | Leu | Leu | Leu | Leu |
| Least frequent AA | | Cys | Cys | Cys | Cys | Cys | Cys | Cys | Cys |
| Parameter | 3rd Base | Number of Codon (TER excluded) | | | | | | | |
| | A/U/C/G | 29 | 29 | 30 | 30 | 29 | 29 | 29 | 29 |
| | A/U | 27 | 27 | 28 | 28 | 27 | 27 | 27 | 27 |
| | G/C | 2 | 2 | 2 | 2 | 2 | 2 | 2 | 2 |
| RSCU > 1 | A | 11 | 11 | 12 | 12 | 11 | 11 | 11 | 11 |
| | U | 16 | 16 | 16 | 16 | 16 | 16 | 16 | 16 |
| | G | 1 | 1 | 1 | 1 | 1 | 1 | 1 | 1 |
| | C | 1 | 1 | 1 | 1 | 1 | 1 | 1 | 1 |
| | A/U/C/G | 20 | 20 | 20 | 20 | 20 | 20 | 20 | 20 |
| | A/U | 18 | 18 | 18 | 18 | 18 | 18 | 18 | 18 |
| | G/C | 2 | 2 | 2 | 2 | 2 | 2 | 2 | 2 |
| Optimal | A | 7 | 7 | 7 | 7 | 7 | 7 | 7 | 7 |
| | U | 11 | 11 | 11 | 11 | 11 | 11 | 11 | 11 |
| | G | 2 | 2 | 2 | 2 | 2 | 2 | 2 | 2 |
| | C | 0 | 0 | 0 | 0 | 0 | 0 | 0 | 0 |

### 3.4. Sequence Divergence and Nucleotide Diversity

To better understand the sequence similarity among the eight species, we use mVista to visualize these differences (Figure 4a). Using *Pl. ovatoauritus* as a reference, we located several high variation regions. They are, regions around the start of the LSC, 21 k, 32 k, 54 k, 64 k, 105 k, region from 12 k to 16.5 k and region from 108 k to 114 k. We can easily infer that the majority of variation happens in two single copy regions while IR regions show a highly conserved characteristic. Some variations are shared commonly while some only exist in the Japanese group. Pi value shows the diversity of the sequence; by using the sliding window method (window length: 600 bp; step size: 200 bp), we obtained the Pi values of different regions (Figure 4b). The Pi values range from 0.00042 to 0.00857. A total of six regions' values are higher than 0.005, and all are located in single copy regions; they are, *trnG-trnT*, *rbcL*, *clpP-psbB*, *psbT-psbH*, *rpl32*, and *ndhI-ndhA*. The ratio of synonymous to nonsynonymous substitutions was calculated to investigate the selecting pressure on nucleotides. We extracted all coding sequences and found that the value of dN/dS reaches 0.565. In the Chinese group, dN/dS reaches 0.613, while in Japanese group, that value decrease to 0.250. To pin down the explicit locations of these polymorphic sites, DNAsp 6.0 was employed. A total of 222 variable sites were detected, including 53 singleton variable

sites and 169 parsimony informative sites. Of all singleton variable sites, 19 sites are located in LSC, 30 sites are located in SSC, 3 sites lie in Ira, and 1 in IRb. Parsimony sites appear predominantly in single copy regions, but LSC has many more sites than SSC (134 sites in LSC while 28 sites in SSC). IRa and IRb own four and three sites separately. The major transversion forms of singleton variable sites are A/T (30.2%) and A/G (26.4%), whereas in parsimony variable sites, A/G and C/T take the dominant place, accounting for 25.4% and 27.2%, respectively (Figure 5).

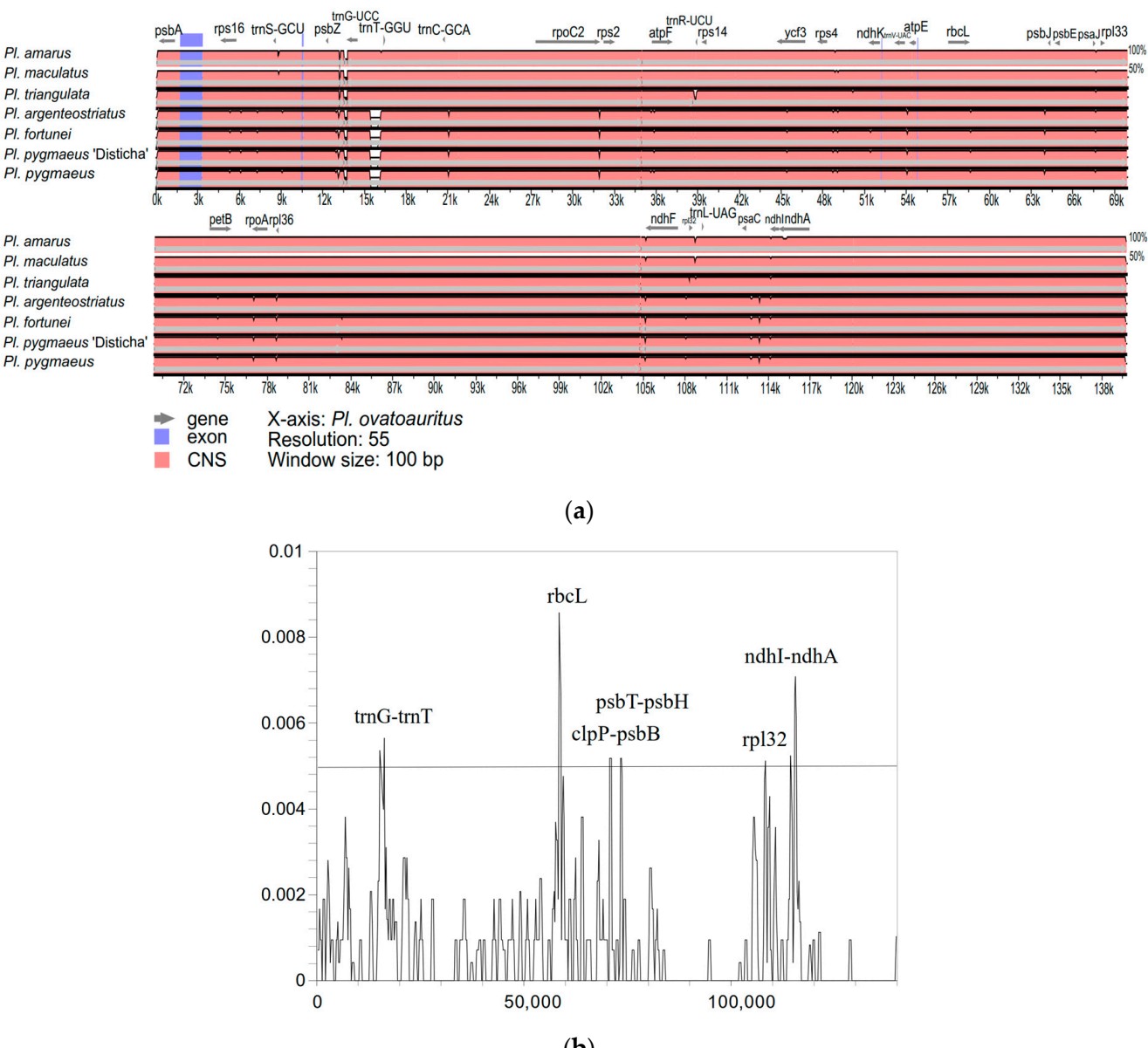

**Figure 4.** (**a**) Sequence divergence analysis using mVista, choosing the Shuffle-LAGAN mode. Blocks in indigo represent exons; red blocks represent none-coding sectors; valleys represent divergences and similarity ranges from 50% to 100%. (**b**) Nucleotide diversity (Pi value) of *Pleioblastus* genus.

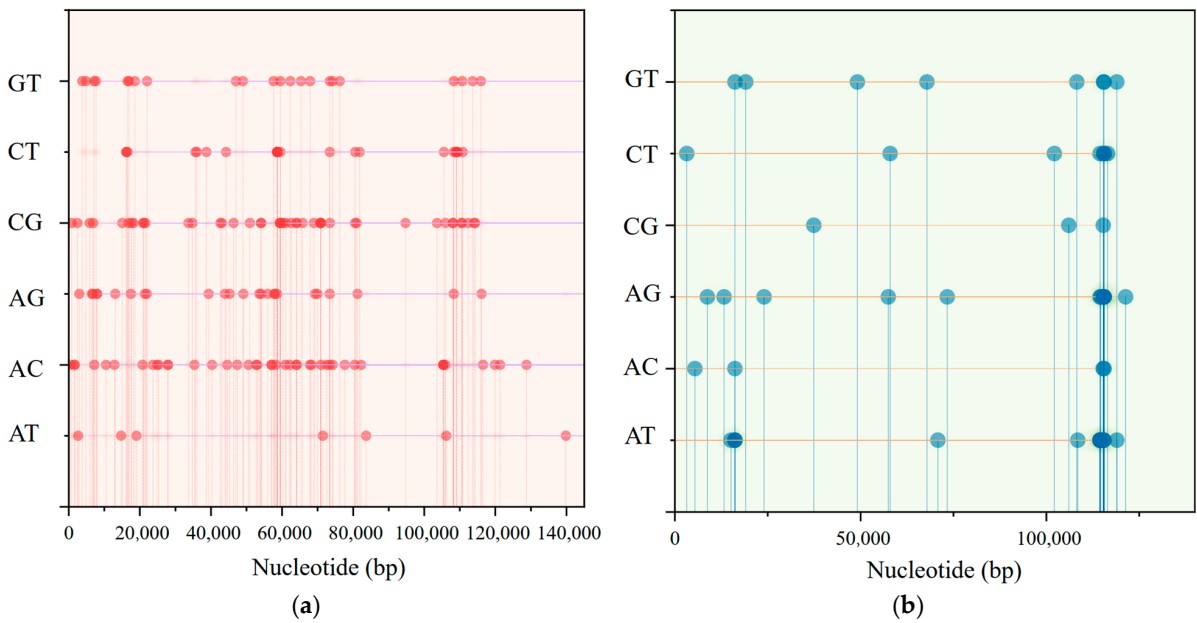

**Figure 5.** Distribution of transversion and transition sites of *Pleioblastus* genus. Horizontal lines represent the base components (i.e., A/T means at a single SNP site, the bases in different species are A or T). (**a**) Red points show the distribution of singleton variable sites; (**b**) blue points show the distribution of parsimony variable sites.

*3.5. Phylogenetic Analysis of Genus Pleioblastus spp.*

In this study, we use the maximum likelihood method to create the phylogenetic tree for 15 species, the result was obtained after 1000 bootstrap replications. When constructing trees, full sequences, protein coding sequences, LSC sequences, SSC sequences as well as IR sequences were used (Figure 6). The result shows high reliability with all confidence levels higher than 98%. According to the ML trees, *Pl. ovatoauritus* is clustered into the Chinese group but also separated from three other species. In most of the trees, *Pl. ovatoauritus* has a closer relationship to *Pl. triangulate*. Different trees showed similar clustering consequences that Chinese and Japanese group are divided, and that the evolutionary distance is shorter in *Pleioblastus* compared to other genera.

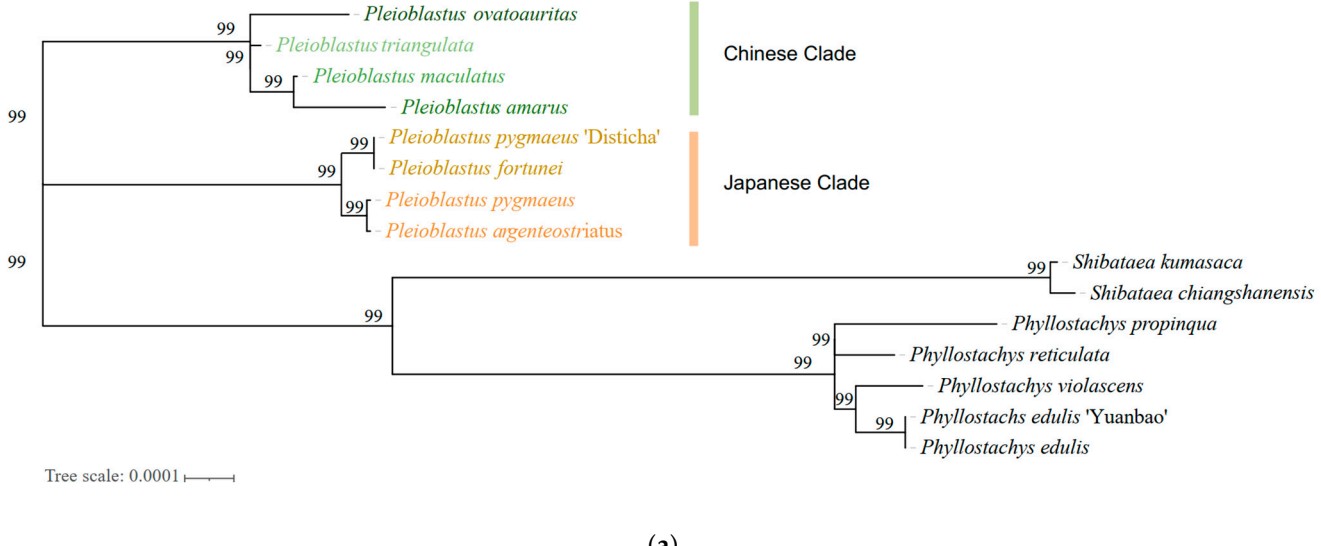

(**a**)

**Figure 6.** *Cont.*

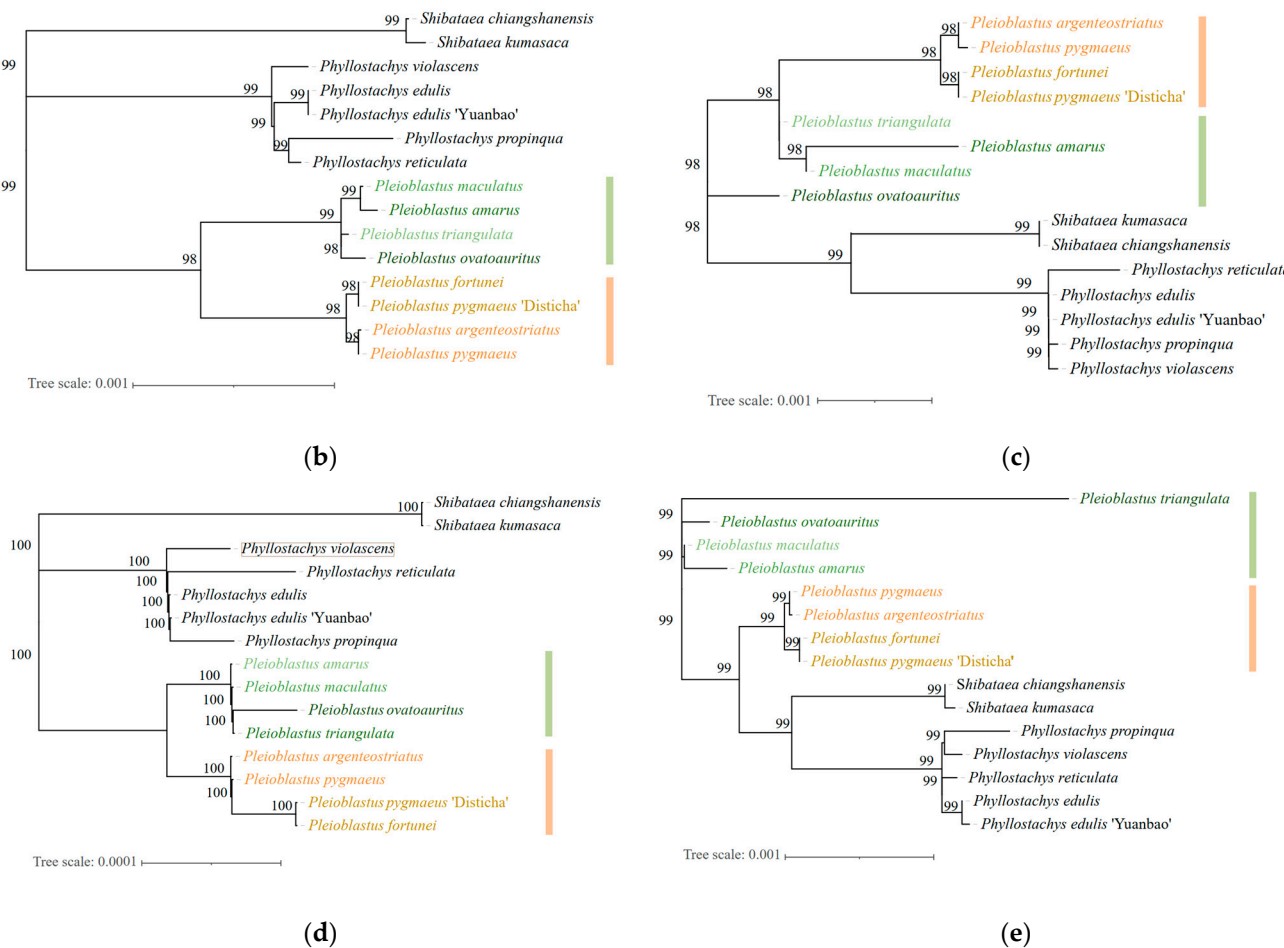

**Figure 6.** The trees with the highest log likelihood are shown based on full-sequence (**a**), LSC sequences (**b**), SSC sequences (**c**), IR sequences (**d**), and protein coding sequences (**e**). The percentage of trees in which the associated taxa clustered together is shown before the branches. Species in the warm color represent the Japanese group, while species in the cold color represent the Chinese group.

*3.6. SSR Distribution and Primer Design of Genus Pleioblastus*

The number and pattern of SSRs varies a lot in different settings. At first, we required the minimum repeat times, reaching 10 for mononucleotide repeats, 6 for dinucleotide repeats, and 5 for the rest. Mono-repeats accounted for all the SSRs in *Pleioblastus,* and the majority of bases are A/T. In the Japanese group, all four species possess 24 SSRs; 23 of them are composited by A/T bases, 1 is composited by G base, and all are located in the LSC region. In the Chinese group *Pl. triangulata* has 27 SSRs, while *Pl. ovatoauritus*, *Pl. maculatus*, and *Pl. amarus* have 25, locating in both LSC and SSC regions. In *Pl. ovatoauritus*, *Pl. amarus*, and *Pl. maculatus*, 23 SSRs are contained in LSC and 2 SSRs are contained in SSC. In *Pl. triangulate*, 26 SSRs are located in LSC and 1 is located in SSC.

When we changed the parameter with lower repeat requirements (nine for mono-, five for di-, and four for the rest), the number boomed and showed an even distribution (Table 5). A/T bases mono-repeats were still the majority, but more G/C bases mono-repeats appeared. With few variants, most repeats have their counterparts located in eight species (Figure 7). Six species had di-repeats, except for *Pl.amarus* and *Pl.maculatus*. All species possessed four trinucleotides repeats in the same location of cp genomes. We designed and developed primers for each SSR and selected nine pairs of high quality (Table A1).

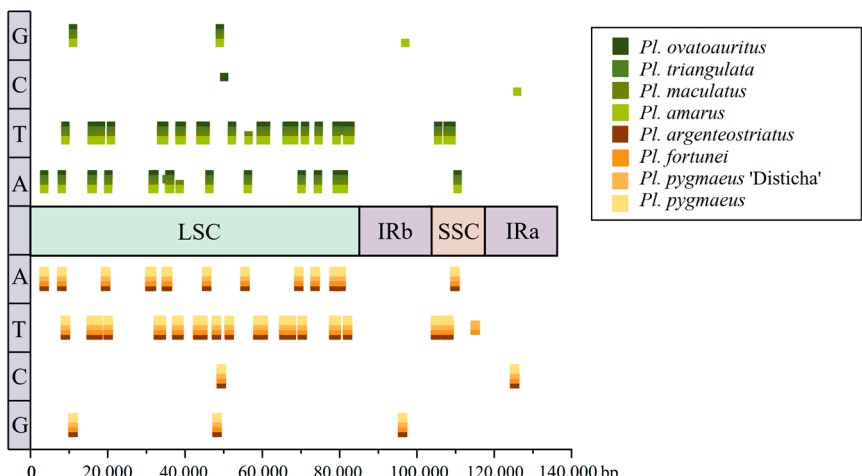

**Figure 7.** Illustration and comparison of mono-SSRs sites in the Chinese group (green) and the Japanese group (orange).

**Table 5.** Number of different types in SSRs 8 *Pleioblastus* spp.

| Motif Length | Base | *Pl. ovatoauritus* | *Pl. triangulata* | *Pl. maculata* | *Pl. amarus* | *Pl. argenteostriatus* | *Pl. fortunei* | *Pl. pygmaeus 'Disticha'* | *Pl. pygmaeus* |
|---|---|---|---|---|---|---|---|---|---|
| Mono-repeats | A | 20 | 20 | 20 | 20 | 18 | 18 | 18 | 18 |
| | T | 29 | 29 | 29 | 29 | 28 | 29 | 29 | 28 |
| | G | 3 | 2 | 2 | 3 | 3 | 3 | 3 | 3 |
| | C | 1 | 0 | 0 | 1 | 2 | 2 | 2 | 2 |
| Total | | 53 | 51 | 51 | 53 | 51 | 52 | 52 | 51 |
| Di-repeats | 5 TA | 2 | 2 | 0 | 0 | 2 | 2 | 2 | 2 |
| | 5 AT | 1 | 1 | 0 | 0 | 1 | 1 | 1 | 1 |
| | 5 TC | 1 | 1 | 0 | 0 | 1 | 1 | 1 | 1 |
| Total | | 4 | 4 | 0 | 0 | 4 | 4 | 4 | 4 |
| Tri-repeats | 4 AAT | 1 | 1 | 1 | 1 | 1 | 1 | 1 | 1 |
| | 4 TAT | 1 | 1 | 1 | 1 | 1 | 1 | 1 | 1 |
| | 4 TCT | 1 | 1 | 1 | 1 | 1 | 1 | 1 | 1 |
| Total | | 3 | 3 | 3 | 3 | 3 | 3 | 3 | 3 |
| Tetra-repeats | 4 GTAG | 1 | 1 | 1 | 1 | 1 | 1 | 1 | 1 |

## 4. Discussion

The chloroplast genome of *Pl. ovatoauritus* was sequenced in this study, yielding a circular genome of 139,708 base pairs with a quadripartite structure similar to that of other bamboo species, such as *Phyllostachys* [32]. Comparative analysis of genome structure, codon usage, sequence divergence, and nucleotide polymorphism were implemented together with its *Pleioblastus* relatives, the results prove that only tiny differences exist among these eight species. This suggests a high level of consistency within the *Pleioblastus* genus compared to other genera. While some annotations of hypothetical genes vary among authors, genes performing basic functions such as photosynthesis, translation, and transcription show a high level of uniformity in number and length. However, differences between the Chinese group and the Japanese group were observed, primarily involving contractions of the boundary in JLB and variations in the distribution and number of SSRs. Further investigation of specific genome regions and individual nucleotides is needed to better understand these differences.

The invasion of the IRa to the *ndhH* gene in the SSC region appears to be a widespread phenomenon in bamboo, with a consistent distance of 187 bp. However, the Chinese group's IRb has invaded more than the Japanese group, with only 142 bp from the JLB to *rps19* gene. Although the number of SSRs differs between species, their locations are consistent within their respective groups, with the Chinese group located in SSC and the Japanese group located in LSC. This distinct distribution provides us with an opportunity to design primers to differentiate these species. While we cannot speculate on which group has a higher evolutionary level or separate these two groups based on these small

differences, these simple clues illustrate that the two groups have undergone explicitly different evolutionary processes. This makes us confident that we can find more solid evidence at the nucleotide level to distinguish the Chinese and Japanese groups.

Using the application mVista, we were able to visualize the divergence and similarity among the eight species and locate several highly variable regions. The most obvious variance is in the region between *trnG*-UCC to *trnT*-GGU, which also showed a high Pi value (over 0.0058). However, this considerable difference is only shared by the Japanese *Pleioblastus* species. Upon examining the aligned sequence, we found a long gap measuring 192 bp, which accounts for this difference. Another notable variance shared by the Japanese group but not caused by long gaps lies in the region of *ndhI-ndhA*. This variable region contributes to the second peak of the Pi value, reaching 0.00708, and the region appears with high single nucleotide polymorphism in both singleton variable sites and parsimony sites, as shown in Figure 4. The summit of Pi comes to 0.00857 in the sector around the *rbcL* gene, while parsimony sites also show great density in this region. However, we could not find any conspicuous variations on the mVista graph. Although some small valleys appear in each species, huge variances together with high Pi values all appear in the four species from Japan. These variances appear with regularity either among the two groups or shared by all eight species, which makes it reasonable to divide the *Pleioblastus* species into two areas.

The phylogenetic results revealed a close relationship among all the *Pleioblastus* species used in comparison to other genera. The newly discovered *Pl. ovatoauritus* is classified in the Chinese group but has a closer evolutionary relationship to *Pl. triangulata* compared to its morphological relative, *Pl. maculatus*. More significantly, among all the maximum likelihood trees, the separation between the Chinese group and the Japanese group is conspicuous. Therefore, as more species are reclassified into the genus *Pleioblastus*, the traditional classification of subgenus or section lacks feasibility. The in-group topological structures of the two clades show high uniformity in all trees. In the Chinese group, *Pl. amarus* and *Pl. maculatus* have a closer relationship, while *Pl. ovatoauritus* is more distant. In the Japanese group, *Pl. argenteostriatus* and *Pl. pygmaeus*, and *Pl. fortunei* and *Pl. pygmaeus* 'Disticha' are clustered, respectively.

In previous studies, based on a four-region cpDNA dataset, Triplett classified *Pl. maculatus* into the Sinicae subclade, while other Japanese *Pleioblastus* species (*Pl. gramineus*, *Pl. pygmaeus*, *Pl. chino*, etc.) were classified into the Medake subclade [33]. Zeng added more data for Chinese species (*Pl. intermedius*, *Pl. juxianensis*, *Pl. amarus*, etc.), but these extra Chinese species did not group with either *Pl. maculatus* or the Japanese species; instead, they were classified into the *Phyllostachys* clade [8]. Zhang concluded that these Chinese species belong to the East China subclade and found them branched into three groups [34]. There seem to be tough problems in resolving the relationships of *Pleioblastus* due to the lack of cp or nuclear genome datasets, both in China and in Japan. At the time of writing this article, we were able to access a total of eight complete cp genomes of *Pleioblastus*, which enabled us to analyze the relationships from a more complex perspective. Researchers working on the phylogeny and genomes of Arundinarieae had found that *Pleioblastus* was not a monophyletic genus, and its distribution in phylograms failed to display the concept of subgenera as depicted in *Flora Reipublicae Popularis Sinicae* [35]. Our results provided evident genomic proof of the accuracy of the dichotomy made for the flora of China.

## 5. Conclusions

The chloroplast genome of *Pl. ovatoauritus* contains 129 genes, including 82 protein-coding genes, 8 rRNAs genes, and 39 tRNAs genes. The codon usage patterns and SSRs sites are highly similar among *Pleioblastus* species. Single nucleotide polymorphism and sequence divergence was found to appear frequently in the single copy region. Six high variable regions (*trnG-trnT*, *rbcL*, *clpP-psbB*, *psbT-psbH*, *rpl32*, and *ndhI-ndhA*) were located. The cluster results indicated that *Pl. ovatoauritus* has a closer relationship to *Pl. triangulata*

and could be classified in the Chinese group. What is more, the explicit split of the Chinese and Japanese group of trees supported the theory that Chinese and Japanese *Pleioblastus* species should be divided into different branches and that the subgenera concept in *Pleioblastus* should be corrected. In the end, we designed nine generic primers of SSRs for *Pleioblastus* with the hope of promoting the germplasm investigation and interspecific relationship analysis of the *Pleioblastus* genus.

**Author Contributions:** Q.G. designed and conducted the research; W.P. conducted data analysis and manuscript writing; B.W. and Z.S. conducted experiments and data analysis. All authors have read and agreed to the published version of the manuscript.

**Funding:** This research was funded by the National Natural Science Foundation of China (31971648) and the Talent Introduction Project Study of Nanjing Forestry University on *Ginkgo biloba* and other important tree germplasm resources (GXL2018001).

**Data Availability Statement:** Data presented in this study are available in the article.

**Acknowledgments:** The authors acknowledge Hu Yaping and Jing Wenxuan from Nanjing Forestry University for software aids, critical reading, and editing of the manuscript.

**Conflicts of Interest:** The authors declare no conflict of interest.

## Appendix A

**Table A1.** Primer designed for *Pleioblastus* based on 8 species.

| Primer | Unit | Forward $5' \rightarrow 3'$ | Reverse $5' \rightarrow 3'$ |
| --- | --- | --- | --- |
| PP1 | (T) 9 | ACCGGTCATGTTTCTTGGAT | AGTCTATTCTCTCTCCTACAACTCT |
| PP2 | (T) 9 | GGCGAACGAATAATCATTAAGTCCT | AGATCCGAACACTTGCCTCG |
| PP3 | (T) 9 | TTCTACGACTCTTTTCCACACT | ATCCAACTGATCCCCACGTC |
| PP4 | T | AAGAAATCGCAACTCCTTTCCG | TCCATGACTCCTATTTCAAAGCCT |
| PP5 | T | TCTCCCCAATAGAGCTTAGAAGT | TCTGGCTGTCTCGCAATACC |
| PP6 | A | CGTGGCTCTAGTATGAATCTAAGGT | TGGCTCATCTGTCTTTCTTTCTT |
| PP7 | AT | TGTGCGTAGAAGAGATTGTGGT | GCTCGAAATGGTTGTGCTCG |
| PP8 | TA | CCCGATCCGATAGTACCCGT | CGTCTTTTGTCATTCTTTGCTCCT |
| PP9 | TA | CGTCTTTTGTCATTCTTTGCTCCT | CCCGATCCGATAGTACCCGT |

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
