# Peer review of "Complete Chloroplast Genome of Bamboo Species Pleioblastus ovatoauritus and Comparative Analysis of Pleioblastus from China and Japan"

_forests, doi:10.3390/f14051051_

Round 1

Reviewer 1 Report

In this manuscript, Peng et al. sequenced the complete plastid genome from a new published bamboo species and conducted comparative analyses in this genus.  The methods used are robust and the results are reliable. I have only minor concern for this manuscript and some grammar errors should be corrected before accepted. For example, line 12-13 change '1', '2' to 'one', 'two'. The sentence in line 57-58 needs reference. In Table 1, P. amarus should be Pl. amarus. Line 139, P. should be Pl.. Line 181, 'gene with copies', should be 'gene with two copies'? 

Author Response

Thank you for carefully reading our manuscript and offering us an opportunity to improve our article (forests-2325109). We appreciated very much the reviewer’s insightful comments and constructive suggestions. All these comments/suggestions were highly valued and addressed in this revision. We hope the revised manuscript has now met the standard of the journal. As required, all revisions have been marked using ‘track changes’ so that you can see them in MS Word.

Reviewer 2 Report

Comments for the Authors

Forests-2325109

Authors did very good work and written very well manuscript entitled “Chloroplast complete genome of new bamboo species Pleioblastus ovatoauritus and comparative analysis of Pleioblastus from China and Japan, together with SSR primer development” but there is some lecuna as given below:

1.      Title should be more concise and informative.

2.      In Abstract, Line no 9, pls rewrite this sentence.

3.       In Abstract, Line no 15, pls rewrite this sentence as you mentioned what is the meaning for that.

4.      Authors need to give final output and outcome of this important experiment and also give final conclusion statement about this study.

5.      In methodology, authors mentioned that plant material collected in 2018 and now they are giving results so are these leaves are collectedin 2018 and stored from last 5 years?

6.      Line no 154, the sentence should be rewrite.

7.      Line no 160, title of table 2 should be rewrite.

8.      Figure 3a should be with high resolution as it is not showing the results clearly.

9.      Figure 4 and 5 should be in high resolution as I can’t read clearly what you want to say from these figures.

10.  Line no 308, P. triangulate should be in italics.

11.  Table 6 should be in supplementary file. I think it is not necessary.

12.  Line no 388, as you mentioned that “our phylogram confirmed the dichotomy of the two nations' species, but the relationships among Subgen. Pleioblastus and Subgen. Nipponocalamus and within the Chinese group remain unclear”. As you say that relationship is unclear but in title you mentioned that you will find out the relationship so how would you conclude the relationship in between two country germplasm?

13.  References should be according to the journal guidelines as Ref no 1, Subtropical Plant Science, should be italics; year should be in bold.

14.  More important English language should be improved.

Author Response

(The authors gave the same response as above.)

Reviewer 3 Report

Dear author, in your work titled "Chloroplast complete genome of new bamboo species Pleio-blastus ovatoauritus and comparative analysis of Pleioblastus from China and Japan, together with SSR primer development" you sequence the chloroplast of a bamboo species and compare it with other ones from the same genus and tribe/subfamily. Then you use all those information to prepare a kit of microsatellites for their differentiation and further research.

Please consider my suggestions as ideas to improve the quality of the manuscript:

Title: as it seems that you are not publishing the existence of this species, I recommend to remove the "new" from the title, although I am not discussing that the description of the species is recent. I would propose:

"Chloroplast complete genome of the bamboo species Pleioblastus ovatoauritus and comparative analysis of the genus Pleioblastus from China and Japan, together with SSR primer development"

-Line 10: P. ovatoauritus instead of Pleioblastus ovatoauritus. Please review in the whole manuscript (i.e. 327, 368).

- Line 15: originated instead of origining (please review the subjunctives from line 10).

-Line 38: "Pleioblastus is a genus of tough woody bamboo" instead of "Pleioblastus is a tough population of woody bamboo"? Population is a word used for individuals from the same species.

- Line 53: There may be a new paragraph describing other species of the subfamily, specially [[ Pl. amarus (Keng) 82 Keng f., Pl. maculatus (McClure) C.D.Chu & C.S.Chao, Pl. triangulata (Hsueh & T.P.Yi) N.H.Xia, Y.H.Tong & Z.Y.Niu, Pl. argenteostriatus (Regel) Nakai, Pl. fortunei (Van Houtte) Nakai, Pl. pygmaeus 'Disticha', Pl. pygmaeus (Miq.) Nakai, Phyllostachys reticu lata (Ruprecht) K. Koch, Ph. edulis (Carriere) J. Houzeau, Ph. edulis 'Yuanbao', Ph. propinqua McClure, Ph. violascens (Carrière) Riviere & C. Rivière, Shibataea chiangshanensis T. W. 87 Wen and S. kumasaca.]] as in M&M they are used as anchoring. The relationships among them should be noted also.

-Line 53: i may suggest to add a picture of the voucher/specimen/natural plant.

-Line 63: species in italics. Please review the italics in manuscript. Also in the subtitles and tables.

-Line 63-64: intraspecific relationshisp of P. ovatoauritus or interespecific relationships within the genus Pleioblastus?

-Line 88: reference to NCBI.

-Line 89: Table instead of Tbale.

-Line 91: Phyllostachys edulis f. tubaeformis instead of Phyllostachys edulis f. tubaeformis. Please review the scientific nomenclatures with f. 

-Line 92: I suggest "The scientific name of the material used in chloroplast genome accession MW874473 , whose name was originally Phyllostachys edulis f. tubaeformis, was corrected to Phyllostachys edulis ‘Yuanbao’, according to the document and confirmation with the author Yue Jin-Jun [15][16]."

instead of

"Notably, according to the document and confirmation with the author Yue Jin-Jun, the scientific name of the material used in chloroplast genome accession MW874473 [15], whose name was originally Phyllostachys edulis f. tubaeformis, was corrected to Phyllostachys edulis ‘Yuanbao’ [16]."

line 103: underline webpage.

line 125: Shibataeta?

line 149: please increase a bit the size of the labels and legend of figure 1

line 160: please add gray horizontal lines between rows (or a smooth alterning colors) as the first column is so narrow that makes the table a bit confusing. Please review the format proposed with the journal rules.

-Line 208-211: please review font size.

-Figure 2, 3 and 5: if possible, please increase a bit the labels, as they are difficult to read in the printed version.

-Table 6: italics for the species name.

-Line 333: genus not italized.

- Line 388-390: please review italics with the general species and sugenus italic rules

REFERENCES:

Please review carefully the style of the references. i.e dates in bold (lines 434, 485). Also the interlining distance of 431-433. 

Author Response

(The authors gave the same response as above.)

Round 2

Reviewer 2 Report

Dear Authors

Thank you for submitting your reply. now paper is in good shape.

In my opinion now paper is in acceptable mode.